# Long-term clinical outcomes after deployment of femoral vascular closure devices in coronary angiography and percutaneous coronary intervention: an observational single-centre registry follow-up

Stephen Wai-Luen Lee, Chor-Cheung Tam, Ka-Lam Wong, Shun-Ling Kong, See-Yue Yung, Yiu-Tung Wong, Suk-Yee Chiu, Cheung-Chi Lam, Ki-Wan Chan, Hon-Wah Chan

Division of Cardiology, Queen Mary Hospital, University of Hong Kong, Hong Kong, Hong Kong

**Correspondence to**
Professor Stephen Wai-Luen Lee; drsl@i-cable.com and prof.stephenlee@gmail.com

## ABSTRACT

**Objectives:** There are few data evaluating the long-term effect of femoral vascular closure devices (FCDs) on patients' clinical outcome. We aim to evaluate the incidence of peripheral vascular disease (PVD) in patients who received FCD following its deployment in coronary angiography and percutaneous coronary intervention (PCI) procedures.

**Design and setting:** Observational study of a single-centre registry.

**Participants:** From June 2000 to September 2004, 265 patients who received FCD after coronary angiography and PCIs were enrolled on the study.

**Outcome measures:** Clinical follow-up (using Rutherford's categories of claudication), ankle brachial index (ABI) and duplex ultrasound of femoral arteries (using the non-accessed side as control) were performed to evaluate the presence of PVD.

**Results:** The mean follow-up interval was 3320 ±628 days. 1 patient (0.4%) suffered from grade 2 claudication and another (0.4%) suffered from grade 1 claudication. The mean ABIs of the accessed side and non-accessed side were 1.06±0.13 and 1.08±0.11, respectively (p=0.17). For duplex ultrasound, the mean common femoral artery peak systolic velocities of the accessed side and non-accessed side were 87.4±22.3 and 87.7±22.1 cm/s, respectively (p=0.73); the mean superficial femoral artery peak systolic velocities of the accessed side and non-accessed side were 81.4±20.1 and 81.31±17.8 cm/s, respectively (p=0.19).

**Conclusions:** The use of FCD after a coronary angiogram and PCI is safe and does not increase the long-term risk of PVD.

## INTRODUCTION

Despite the increasing trend of utilising radial access for a coronary angiogram and

### Strengths and limitations of this study

- This study showed no increase in incidence of clinically significant peripheral vascular disease after use of femoral vascular closure device (FCD) in long-term follow-up (median follow-up time approaching 10 years).
- The use of duplex ultrasound showed no evidence of adverse effect of FCD on the accessed femoral artery.
- The lack of baseline ultrasound before vascular closure device precluded precise evaluation of effect of vascular remodelling of the femoral artery after collagen plug and suture deployment.
- It may not be possible to generalise the current study results to other vascular closure devices or in settings of large size femoral access as in structural heart disease intervention.

percutaneous coronary intervention (PCI), femoral artery access haemostasis is still an integral part in the field of interventional cardiology. Femoral vascular closure devices (FCDs) are designed specifically to facilitate arteriotomy closure, which has been shown to allow for a shorter haemostasis time period and early ambulation.[1–3] However, there are no data to demonstrate the superiority of FCD in reducing vascular complications when compared with manual compression,[1 4 5] and the long-term effect of FCD on peripheral vasculature has not been extensively investigated, apart from a scanty case report describing the occurrence of delayed claudication post-FCD use.[6 7] Particularly, the impingement of a femoral artery by FCD with subsequent inflammation

and remodelling can take place for years, which potentially causes vascular stenosis. Our group has previously published data[8] on the in-hospital and 1-year clinical outcomes in patients who received FCD after coronary angiogram and PCIs earlier, and we performed this follow-up study to evaluate the incidence of lower limb peripheral vascular disease (PVD) in the same group of patients in the long run.

## METHODS

The design of the study has been described elsewhere.[8] In brief, from June 2000 to September 2004, 265 patients who successfully received FCDs after a coronary angiogram and PCIs were enrolled for the study. A femoral arteriogram was routinely performed before consideration of arterial closure, and two types of FCD namely Angio-Seal (St Jude Medical, Minnesota, USA) and Perclose (Abbott, Abbott Park, Illinois, USA) closure devices were used. Angio-Seal is a collagen-mediated device which utilises a bioabsorbable collagen sponge to seal the arteriotomy site, while Perclose achieves haemostasis by delivering a knot to suture the arteriotomy.

For the long-term registry follow-up, all patients were seen in our clinic every 4–6 months and occurrence of any vascular complications was recorded. Clinical symptoms and signs of lower limb ischaemia were assessed by Rutherford's grade of claudication,[9] and ankle brachial index (ABI) was measured by using a sphygmomanometer and vascular Doppler probe (Hadeco MiniDop ES-100VX). Patients were instructed to lie comfortably on a bed and a blood pressure cuff was applied to the patients' arms; a Doppler probe was positioned at the brachial artery to obtain the systolic brachial pressure while the cuff was then placed at the calves with the Doppler probe positioned at the dorsalis pedis or posterior tibial artery (whichever is higher) to obtain the ankle systolic pressure. In addition, duplex ultrasound using a high-frequency linear array transducer was performed to assess the peak systolic velocity at the common femoral artery (CFA) and superficial femoral artery (SFA). The non-accessed side was used as the control to compare values of ABI and peak systolic velocity to the side where FCDs were deployed.

Continuous variables were expressed as the mean and SD. Dichotomous variables were expressed as counts and percentage. Statistical comparisons were performed using the Student t test or Wilcoxon signed-rank test for continuous variables as appropriate while Fisher's exact test was used for categorical variables. A p value <0.05 was considered statistically significant. All statistical analyses were performed using SPSS for Windows (V.19.0, SPSS Inc, Chicago, Illinois, USA).

## RESULTS

Two hundred and sixty-five patients were entered in the study; the baseline clinical and procedural characteristics were shown in table 1. The mean age of patients was $61.8\pm10.4$ years with 80% being men. Vascular closure devices were used in 235 PCI (88.7%) and 30 (11.3%) coronary angiographic procedures. Sixty procedures (22.6%) used Angio-Seal, whereas the Perclose closure device was deployed in 205 procedures (77.4%). Most of the vascular closure devices used were 6 F in size. Patients continued to be followed up, and for the current study, 233 patients were recruited for analysis while ABI and duplex ultrasound studies were performed in 145 patients (figure 1).

The mean follow-up time was $3320\pm628$ days. Out of 233 patients with long-term follow-up, there was one case (0.4%) of Rutherford's grade 2 claudication with a 79-year-old woman with end-stage renal failure and bilateral calf claudication associated with rest pain. There was another case (0.4%) of bilateral Rutherford's grade 1 claudication in a 78-year-old man with bilateral lower limb PVD, and duplex ultrasound confirmed right SFA occlusion and left CFA 70% stenosis. The remaining patients were free of symptomatic PVD (table 2).

Table 2 also showed the ABI and duplex ultrasound results on long-term follow-up. The ABIs of accessed and non-accessed sites were $1.06\pm0.13$ and $1.08\pm0.11$,

| Table 1 | Baseline characteristics | | |
|---|---|---|---|
| **Clinical characteristics** | | **Procedural variables** | |
| Mean age (years) | 61.8±10.4 | Coronary angiogram, n (%) | 30 (11.3) |
| Gender, male (%) | 213 (80.4) | PCI, n (%) | 235 (88.7) |
| Current smoker, n (%) | 54 (20.4) | Angio-Seal, n (%) | 60 (22.6) |
| Diabetes mellitus, n (%) | 63 (23.8) | Perclose, n (%) | 205 (77.4) |
| Hypertension, n (%) | 130 (49.1) | Sheath size | |
| Hyperlipidaemia, n (%) | 147 (55.5) | 6 F, n (%) | 237 (89.4) |
| Peripheral vascular disease, n (%) | 8 (3.0) | 7 F, n (%) | 2 (0.8) |
| | | 8 F, n (%) | 26 (9.8) |
| Renal failure, n (%) | 7 (2.6) | Mean ACT, s | 346±61 |
| Left ventricular function (%) | 64.9±14.3 | Mean procedural time for | 147±121 |
| Body mass index, kg/m$^2$ | 24.9±3.1 | achieving haemostasis, s | |
| | | Right side:left side | 256:7 |

ACT, activated clotting time; PCI, percutaneous coronary intervention.

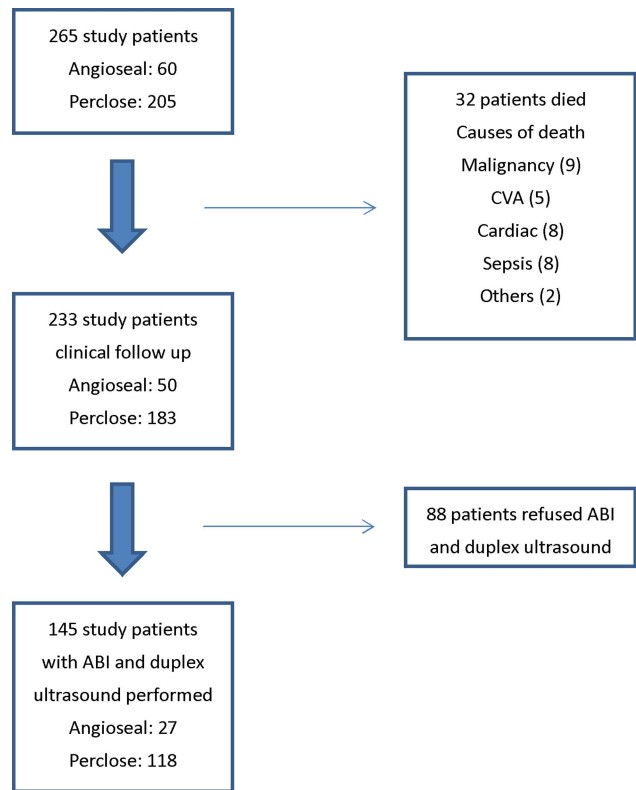

**Figure 1** Study patients. ABI, ankle brachial index; CVA, cerebrovascular accident.

respectively. Duplex ultrasound measured peak systolic velocities of CFA in accessed and non-accessed sites were 87.43±22.34 and 87.69±22.06 cm/s, respectively, and of SFA were 81.40±20.85 and 81.31±17.80 cm/s, respectively. There was no statistically significant difference in the above parameters between the accessed and non-accessed sites. Nevertheless, the mean end-diastolic diameter (EDD) of CFA and SFA in accessed sites was significantly larger than that in non-accessed sites (CFA: 8.12±1.27 vs 7.80±1.16 mm, p<0.01; SFA: 6.58±0.96 vs 6.38±0.95 mm, p<0.01).

Data were also analysed between the Angio-Seal and Perclose subgroups and results were shown in table 3.

Baseline clinical and procedural characteristics were similar between both groups while patients who received Perclose had a non-significantly higher peak systolic velocity in CFA and SFA. The mean EDD of CFA in accessed sites was significantly larger in patients who received the Perclose device than those who received Angio-Seal (Perclose: 8.25±1.28 mm vs Angio-Seal: 7.58±1.10 mm, p=0.02), but it is not the case for mean EDD of SFA.

## DISCUSSION

To the best of our knowledge, this is the first study to evaluate the long-term impact of FCD on incidence of lower limb PVD. In the study published earlier, we demonstrated the favourable in-hospital and 1-year outcome of usage of FCD in our patients while this study showed no evidence of increase in risk of lower limb PVD after a mean follow-up of almost 10 years. Most of the previous studies including meta-analysis[1–5] comparing FCD and manual compression emphasised on evaluating short-term vascular complications as a safety outcome. Theoretically, use of a collagen plug and suture-mediated vascular closure on an arteriotomy site can impinge on the CFA causing inflammation, remodelling and flow disturbances which potentially accelerate atherosclerosis. The process can take years to result in clinically significant artery narrowing. Our study used the non-accessed side as the control and it identified nil increased incidence of clinical PVD by symptom, ABI and duplex ultrasound. Two patients suffering from symptomatic PVD had their right femoral artery accessed previously, and the occurrence of bilateral lower limb arterial disease was unlikely to be accounted for by use of FCD. On the contrary, the EDD of CFA and SFA in accessed sites in our patients was larger than that in non-accessed sites. Since we did not have baseline ultrasound of bilateral femoral arteries before device deployment, we cannot ascertain whether the discrepancy is due to nature difference or device-related injury and positive remodelling. It has been shown in a study

| Table 2 | Follow-up results | | |
|---|---|---|---|
| **Clinical outcomes** | | | |
| Rutherford's grade of claudication | | | |
| Grade 0, n (%) | 231 (99.2) | | |
| Grade 1, n (%) | 1 (0.4) | | |
| Grade 2, n (%) | 1 (0.4) | | |
| | **Access site** | **Control site** | **p Value** |
| Ankle brachial index | 1.06±0.13 | 1.08±0.11 | 0.17 |
| **Duplex ultrasound** | **Access site** | **Control site** | **p Value** |
| CFA peak systolic velocity, cm/s | 87.43±22.34 | 87.69±22.06 | 0.73 |
| SFA peak systolic velocity, cm/s | 81.40±20.85 | 81.31±17.80 | 0.19 |
| CFA end-diastolic diameter, cm | 8.12±1.27 | 7.80±1.16 | <0.01 |
| SFA end-diastolic diameter, cm | 6.58±0.96 | 6.38±0.95 | <0.01 |
| ABI, ankle brachial index; CFA, common femoral artery; SFA, superficial femoral artery. | | | |

**Table 3** Angio-Seal versus Perclose

| | Angio-Seal (n=60) | Perclose (n=205) | p Value |
|---|---|---|---|
| Baseline characteristics | | | |
| Mean age (years) | 62.4±10.8 | 61.7±10.3 | 0.67 |
| Gender, male (%) | 50 (83.3) | 162 (79.8) | 0.71 |
| Diabetes mellitus, n (%) | 19 (31.7) | 44 (21.7) | 0.12 |
| Hypertension, n (%) | 27 (45.0) | 101 (49.8) | 0.56 |
| Body mass index, kg/m$^2$ | 24.6±3.4 | 24.9±3.1 | 0.53 |
| Coronary angiogram, n (%) | 5 (8.5) | 25 (12.5) | 0.49 |
| PCI, n (%) | 55 (91.5) | 178 (87.5) | 0.49 |
| Parameters on accessed side | | | |
| Ankle brachial index | 1.08±0.09 | 1.06±0.14 | 0.98 |
| CFA peak systolic velocity, cm/s | 84.66±15.09 | 88.07±23.70 | 0.55 |
| SFA peak systolic velocity, cm/s | 80.95±21.10 | 81.50±20.89 | 0.79 |
| CFA end-diastolic diameter, cm | 7.58±1.10 | 8.25±1.28 | 0.02 |
| SFA end-diastolic diameter, cm | 6.37±1.09 | 6.63±0.93 | 0.25 |

ABI, ankle brachial index; CFA, common femoral artery; PCI, percutaneous coronary intervention; SFA, superficial femoral artery.

that the diameter of the right CFA is larger than that of the left CFA,[10] with a mean difference of 0.26 mm, which is similar to the difference of 0.32 mm between our patients' accessed (>99% right side) and non-accessed sites. Further studies are needed to verify whether the femoral closure device will have an impact on femoral artery positive remodelling.

In the animal model,[11] the collagen plug produced a more intense inflammatory reaction of the tissue surrounding the femoral arteries when compared with the suture-mediated device, but this did not lead to a higher incidence of clinical PVD and duplex ultrasound-derived flow abnormalities in our patients who received Angio-Seal. On the other hand, the diameter of the accessed CFA is larger in Perclose than in Angio-Seal. We cannot determine whether a more intense inflammation with Angio-Seal gives rise to a smaller vessel calibre in the long run or Perclose triggers a more pronounced positive remodelling. Nonetheless, this interesting finding that highlights tissue remodelling may differ between different mechanisms of vascular healing.

## LIMITATIONS
We did not have baseline duplex ultrasound in our patients, and we used the non-accessed side as the control for which we assumed the baseline appearance of both sides of the femoral arteries as identical. In addition, this was a single-centre study and the sample size was relatively small, which may be underpowered to detect rare vascular events. Furthermore, more than one-third of the patients did not consent for ABI and duplex ultrasound measurement, making the data less representative. Moreover, FCDs used in the current study were either Angio-Seal or Perclose from 6 to 8 F, and hence it may not be possible to generalise the results to other FCDs or in settings of larger size femoral access as in structural heart disease intervention.

## CONCLUSIONS
Use of FCDs was safe and did not increase the incidence of lower limb PVD on long-term follow-up.

**Contributors** SW-LL initiated the concept of work, performed the analyses in conjunction with the coauthors and wrote and revised the manuscript. C-CT, K-LW, S-LK, S-YY, Y-TW and S-YC performed clinical and ultrasound assessment. All authors contributed to the revisions of the manuscript and the interpretation of the findings.

**Funding** This research received no specific grant from any funding agency in the public, commercial or not-for-profit sectors.

**Competing interests** None.

**Ethics approval** Institutional review board of the university of Hong Kong, Hospital Authority Hong Kong West Cluster.

**Provenance and peer review** Not commissioned; externally peer reviewed.

**Data sharing statement** No additional data are available.

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
