## [Reviewer comments · BMJ Open]

Some articles will have been accepted based in part or entirely on reviews undertaken for other BMJ Group journals. These will be reproduced where possible.

ARTICLE DETAILS

TITLE (PROVISIONAL)	Long Term Clinical Outcomes After Deployment of Femoral Vascular Closure Devices in Coronary Angiography and Percutaneous Coronary Intervention – An observational single center registry follow up
AUTHORS	Lee, Wai Luen Stephen; TAM, Frankie; WONG, Michael; KONG, Shun-Ling; YUNG, Arthur; WONG, Anthony; Chiu, SY; LAM, Simon; CHAN, Kelvin; CHAN, Raymond

VERSION 1 - REVIEW

REVIEWER	George Dangas Mount Sinai, USA
REVIEW RETURNED	05-Apr-2014

GENERAL COMMENTS	In the introduction section the background is poor (lacking related references) while it is not clear why the authors decided to access the long term outcomes by using ultrasound and ABI measurements. In the Methods section the authors should describe even in brief the study design. Also they need to mention the closure devices that have been used along with a short description. The reference for the Rutherford classification should be added. The Discussion section does not explain, justify and support the results. It should be re-written in a more comprehensive way. In the Figure the authors should add in each box the number and the type of the devices that were used.
---

REVIEWER	Fausto Biancar Department of Surgery, Oulu University Hospital, Oulu, Finland.
REVIEW RETURNED	10-Apr-2014

GENERAL COMMENTS	The authors reported the results of a study assessing the safety of vascular closure devices in patients undergoing coronary angiography and intervention. The article is rather well written and data nicely presented. I have a few comments: 1. The authors published in 2010 the intermediate results of this study, which indeed are similar to the present ones. I am not sure it sound to report these data twice.2. Page 6, line 10: change "femoral artery wound" to "femoral artery access".3. Page 7, line 41: please write "the mean and standard deviation".4. Is this a prospective study?
--

	5. Why did so many patients not accept ultrasound examination? 6. Page 9: were these cases of lower limb claudication related to the use of VCD? This is not clear.
--	---

VERSION 1 – AUTHOR RESPONSE

Response to Reviewer 1:

1) In the introduction section the background is poor (lacking related references) while it is not clear why the authors decided to assess the long term outcomes by using ultrasound and ABI measurements.

The reviewer is right to point out these points. The whole point of the current study is to investigate whether the use of FCD will adversely affect lower limb vasculature and increase the incidence of lower limb PVD. We use ultrasound and ABI as they are of high sensitivity, specificity and non-invasive to detect subclinical PVD. The introduction part is modified with appropriate references cited.

2) In the Methods section the authors should describe even in brief the study design. Also they need to mention the closure devices that have been used along with a short description.

We apologize for this confusion and we have tried to make our manuscript over-brief. The study design section has been modified and short descriptions of FCD (Angioseal and Perclose) are added.

3) The reference for the Rutherford classification should be added.

Thanks for pointing out this. Appropriate reference cited.

4) The Discussion section does not explain, justify and support the results. It should be re-written in a more comprehensive way.

Thanks for the effort to improve our manuscript. The discussion section has been rewritten to try to explain the results of our analysis. In brief, there is no increase in incidence of lower limb PVD in 10 year follow up by means of 1. clinical symptom and sign; 2. ABI and 3. Duplex ultrasound.

Besides, we have been trying to explain the discrepancies of common femoral artery diameter difference between accessed and non-accessed sides and also between angioseal and perclose group. We acknowledge the lack of baseline ultrasound is a major limitation to reach a solid conclusion and we hope it sheds light on any future research on this topic.

5) In the Figure the authors should add in each box the number and the type of the devices that were used.

Comments noted and figure modified.

Response to reviewer 2:

1) The authors published in 2010 the intermediate results of this study, which indeed are similar to the present ones. I am not sure it sound to report these data twice.

Thanks for the comment and we agree the results of current study is similar to previous one.

However, what makes the new data important is the long follow up time (a median of 10 years). The previous study published 1 year outcome data and we understand the effect of vascular remodeling takes years to become apparent. This point has also been raised by Professor Zoltan G. Turi in the

editorial of our previous publication (Catheterization and Cardiovascular Interventions 2010 75:349-350). We hope the current analysis confirm our hypothesis that FCD does not have adverse consequence on peripheral vasculature or at least does not increase incidence of PVD in the long run.

2) Page 6, line 10: change "femoral artery wound" to "femoral artery access".

Noted and modified.

3) Page 7, line 41: please write "the mean and standard deviation".

Thanks for pointing out this. Content has been changed accordingly.

4) Is this a prospective study?

The reviewer is correct to highlight this point as the editor has the identical concern. We consider it will be more appropriate to regard this study as an observational study of a single center registry. Appropriate amendments in title, abstract and methods sections have been made.

5) Why did so many patients not accept ultrasound examination?

We are also disappointed by the substantial refusal rate of performing ABI and ultrasound. Patients were keen to come back for clinical follow up. However, the ABI and ultrasound machines were at a different site in our institution. Not surprisingly a number of patients declined to go a less familiar location as an extra visit. We must also express our gratitude to the remaining patients who spend their time to travel and participate in the study.

6) Page 9: were these cases of lower limb claudication related to the use of VCD? This is not clear.

We shared the same view with the referee that it is not clear whether the use of FCD lead to clinical lower limb claudication in the two patients. From our opinion, those two patients with vascular risk factors suffered from bilateral PVD and they both have their right common femoral artery accessed by FCD only. It is unlikely their symptoms can be accounted by FCD use and this point has also been added to the discussion section.